# A novel ultrasound-guided mouse model of sudden cardiac arrest

**Cody A. Rutledge[1], Takuto Chiba[2,3], Kevin Redding[1], Cameron Dezfulian[4], Sunder Sims-Lucas[2,3], Brett A. Kaufman**  **[1] ***

**1** Division of Cardiology, Cardiovascular Institute, University of Pittsburgh, Pittsburgh, PA, United States of America, **2** Rangos Research Center, Children's Hospital of Pittsburgh, University of Pittsburgh, Pittsburgh, PA, United States of America, **3** Division of Nephrology, Department of Pediatrics, University of Pittsburgh School of Medicine, Pittsburgh, PA, United States of America, **4** Safar Center for Resuscitation Research and Critical Care Medicine Department, University of Pittsburgh, Pittsburgh, PA, United States of America

* bkauf@pitt.edu

**Data Availability Statement:** The data underlying the results presented in the study are available from OSF at https://osf.io/f7ayv/.

**Funding:** Research reported in this manuscript was supported by: American Heart Association

## Abstract

### Aim

Mouse models of sudden cardiac arrest are limited by challenges with surgical technique and obtaining reliable venous access. To overcome this limitation, we sought to develop a simplified method in the mouse that uses ultrasound-guided injection of potassium chloride directly into the heart.

### Methods

Potassium chloride was delivered directly into the left ventricular cavity under ultrasound guidance in intubated mice, resulting in immediate asystole. Mice were resuscitated with injection of epinephrine and manual chest compressions and evaluated for survival, body temperature, cardiac function, kidney damage, and diffuse tissue injury.

### Results

The direct injection sudden cardiac arrest model causes rapid asystole with high surgical survival rates and short surgical duration. Sudden cardiac arrest mice with 8-min of asystole have significant cardiac dysfunction at 24 hours and high lethality within the first seven days, where after cardiac function begins to improve. Sudden cardiac arrest mice have secondary organ damage, including significant kidney injury but no significant change to neurologic function.

### Conclusions

Ultrasound-guided direct injection of potassium chloride allows for rapid and reliable cardiac arrest in the mouse that mirrors human pathology without the need for intravenous access. This technique will improve investigators' ability to study the mechanisms underlying post-arrest changes in a mouse model.

Transformation Project Award 18TPA34230048 (to BK), NIH Instrument Grant for Advanced High-Resolution Rodent Ultrasound Imaging System 1S10OD023684-01A1 (to Dr. Kang Kim, University of Pittsburgh), NIH Training Program in Imaging Sciences in Translational Cardiovascular Research 5T32HL129964-02 (to CR), and Richard K. Mellon Institute Award for postdoctoral trainees (to TC). The funders had no role in study design, data collection and analysis, decision to publish, or preparation of the manuscript.

**Competing interests:** The authors have declared that no competing interests exist.

## Introduction

Out of hospital cardiac arrest affects over 350,000 patients annually in the United States [1]. Only 10% of these patients will survive to hospital discharge and only 6% of the patients will be discharged with a favorable outcome [2, 3]. These statistics suggest a dramatic need for novel interventions to improve cardiac arrest outcomes. Numerous animal models of sudden cardiac arrest (SCA) have been developed to help understand the mechanisms underlying cardiac arrest mortality and to explore potential interventions in pre-clinical models [4, 5]. A review of animal models of SCA found that only 6% of pre-clinical SCA studies were completed in mice [5]. A mouse model has a number of advantages over large animal models, including rapid breeding, cost-effectiveness, and opportunity for genetic manipulation [6]. However, mouse use to model SCA has been limited by the time required to obtain venous access and difficulty delivering life support related to the animal's size [7, 8].

The major limitation to adoption of the mouse model is the difficult cannulation of the delicate femoral or internal jugular (IJ) veins, resulting in prolonged surgical times and high lethality necessitating extensive training in the model [9]. In this paper, we describe a novel method for ultrasound-delivery of potassium chloride (KCl) directly into the left ventricle (LV) to induce immediate cardiac arrest, bypassing the need for establishing intravenous access. This direct cardiac injection method simplifies the surgical process Our model has short procedure times and high rate of surgical survival, thereby increasing the opportunity for study of preventative and therapeutic interventions, as well as mechanistic aspects of organ damage and recovery.

## Material and methods

### Animal preparation

8-week-old C57BL/6J male and female mice were anesthetized using 5% isoflurane in 100% oxygen via induction box until reaching the surgical plane. These mice were placed in a supine position and quickly intubated while anesthetized by placing a 22 g catheter endotracheally and then mechanically ventilated (MiniVent, Harvard Apparatus, Holliston, MA) at a rate of 150 bpm (125 μL for females and 140 μL for males). Animal temperature was maintained near 37˚C via heating pad and rectal temperature probe and heart rate (HR) was monitored continuously using surface electrocardiogram (ECG; Visual Sonics, Toronto, Canada). HR was maintained between 400–500 bpm by adjusting isoflurane concentration. Depilatory cream was applied to the thorax and the chest cleaned with alcohol. All studies were performed at the University of Pittsburgh in compliance with the National Institutes of Health Guidance for Care and Use of Experimental Animals and was approved by the University of Pittsburgh Animal Care and Use Committee (Protocol #18032212).

### Ultrasound-guided KCl delivery and cardiopulmonary resuscitation (CPR)

Baseline transthoracic echocardiography was performed using the Vevo 3100 imaging systems (Visual Sonics) with a 40 MHz linear probe along the long-axis of the heart. A 30-gauge needle was carefully advanced under ultrasound-guidance through the intercostal space and directed into the LV. Occasional premature ventricular contractions were noted during needle puncture, typically 1–3 beats, but no sustained arrhythmias were recorded. 40 μL of 0.5M KCl in saline (pre-warmed to 37˚C) was delivered into the LV cavity, causing immediate asystole as observed on lead 2 of surface ECG. The ventilator was turned off at this time. Doppler imaging was utilized over the aortic outflow tract to confirm that no blood was ejected during asystole. The mice remained in asystole for 7.5 minutes, when a second 30-gauge needle was introduced

into the LV and 500 μL of 15 μg/mL epinephrine in saline (37°C) was delivered over approximately 30 seconds. At 8 minutes, the ventilation was resumed at 180 bpm and CPR initiated. CPR was performed manually by finger compressions just lateral of the xiphoid process, aiming for a depth of 1/3 to 1/2 of the animal's thorax, at 300 bpm for 1 min. At 1 min, CPR was briefly held and ECG evaluated for recovery of spontaneous circulation (ROSC). If a sinus rhythm was observed on ECG, doppler imaging was performed to confirm aortic blood flow. If not, one to two additional 1-minute cycles of CPR were performed. Animals not achieving ROSC by 3 minutes were euthanized. Mice remained on the ventilator (without isoflurane) for approximately 20–25 minutes until breathing spontaneously at a rate over 60 respirations/ minute. Sham mice received no KCl injection. Rather, sham animals were treated with a single, direct LV injection of 500 μL epinephrine in normal saline. Sham animals did not experience asystole or receive chest compressions, and were extubated within minutes after injection. All animals were placed in a recovery cage under heat lamp for 2 hours with hourly temperature monitoring by rectal thermometer for up to 4 hours.

## Survival analysis

A cohort of sham and SCA mice that survived the initial surgery were designated for survival analysis over a 4-week time frame. 5 sham mice and 13 SCA mice were included in this study. Survival was assessed every morning over 4-weeks post-operatively.

## Ultrasound and echocardiography

Echocardiography was performed at baseline, 1 day, 1 week, and 4 weeks as previously described [10] and measured by a blinded operator. Briefly, transthoracic echocardiography was performed using the Vevo 3100 system and analyzed using VevoLab v3.2.5 (Visual Sonics). B-mode images were taken for at least 10 cardiac cycles along the parasternal long axis of the LV and end-systolic volume (ESV) and end-diastolic volumes (EDV) calculated by modified Simpson's monoplane method [11]. Short-axis M-mode images were obtained at the level of the papillary muscle for representative images only. Ejection Fraction (EF) was calculated from long-axis B-mode imaging as: $100 \times (LV\ EDV - LV\ ESV) / (LV\ EDV)$.

A cohort of mice were assessed for renal perfusion following ROSC. The ultrasound probe was oriented transversely across the abdomen at the plane of the right kidney. Mice remained unconscious for approximately one hour after SCA, allowing for ultrasound without the need for additional anesthesia. Doppler imaging over the renal artery evaluated the presence of blood flow every thirty seconds until sustained blood flow was noted.

## Serum analysis

After euthanasia, mice underwent cardiac puncture for collection of blood by heparinized syringe. Blood was separated by centrifugation at 2000 x g at 4°C for 10 minutes and the serum was flash frozen. These samples were evaluated for blood urea nitrogen (BUN), serum creatinine, alanine aminotransferase (ALT), and creatine kinase (CK) by the Kansas State Veterinary Diagnostic Laboratories (Manhattan, KS).

## Tissue histology

Kidneys were fixed overnight in 10% formalin at 4°C then washed with PBS and transferred to 70% ethanol at room temperature. After fixation, tissues were embedded into paraffin prior to sectioning at 4 microns by the Histology Core at the Children's Hospital of Pittsburgh. Sections were stained with hematoxylin and eosin (H&E). Renal tubular pathology was semi-

quantitatively scored (0: no injury to 4: severe injury) in terms of tubular dilatation, formation of proteinaceous casts, and loss of brush border [12]. Histological scoring was performed in a blinded fashion at 40x magnification on outer medullary regions of the tissue sections. Eight fields were evaluated per sample. Samples were imaged using a Leica DM 2500 microscope (Leica, Wetzlar, Germany) and LAS X software (Leica).

## Statistical analysis

Data were expressed and mean ± standard error in all figures. $p \leq 0.05$ was considered significant for all comparisons. To determine whether sample data has been drawn from a normally distributed population, D'Agostino-Pearson test was performed. For parametric data, Student's t-test was used to compare two different groups. For nonparametric data, Mann-Whitney test was used. Survival analysis was assessed by using Kaplan-Meier and log rank (Mantel-Cox) testing. All statistical analysis was completed using Graphpad Prism 8 software (San Diego, CA).

## Results

### Baseline sex, weight, EF and HR are similar between groups

19 sham and 30 arrest mice were evaluated in this study. There was no difference in baseline weight (sham: 22.6±0.9 g; arrest: 23.1±0.6 g, Table 1), EF (sham: 59.8±1.5%; arrest: 59.9±1.0%) or HR (sham: 472±19 bpm; arrest: 488±11 bpm) between groups. A single mouse from the sham group (1/19, 5.2%) died immediately following extubation and five mice died from the arrest group (5/30, 16.7%), where four mice did not achieve ROSC and one mouse died

**Table 1. Physiologic and surgical characteristics of Sham and SCA Mice.**

|  | Sham (±SEM) | Arrest | p-value |
|---|---|---|---|
| Age (d) | 56.8±0.7 (n = 18) | 57.7±0.6 (n = 25) | 0.34 |
| Weight (g) | 22.6±0.9 (n = 15) | 23.1±0.6 (n = 25) | 0.63 |
| Sex | 10 female, 9 male | 14 female, 16 male | n/a |
| Surgical Survival |  |  |  |
| Total | 18/19 | 25/30 | n/a |
| Males | 9/9 | 14/16 |  |
| Females | 9/10 | 11/14 |  |
| CPR Duration | n/a | 1.32±0.11 (n = 25) | n/a |
| Time to Extubation | n/a | 22.7±0.7 (n = 25) | n/a |
| Initial Body Temp (˚C) | 35.7±0.2 (n = 15) | 35.5±0.2 (n = 25) | 0.79 |
| ROSC Body Temp (˚C) | n/a | 35.2±0.2 (n = 25) | n/a |
| 1 h Body Temp (˚C) | 35.9±0.1 (n = 7) | 35.9±0.3 (n = 14) | 0.95 |
| 2 h Body Temp (˚C) | 35.8±0.1 (n = 7) | 35.2±0.3 (n = 13) | 0.18 |
| 3 h Body Temp (˚C) | 35.8±0.1 (n = 6) | 33.0±0.6 (n = 9) | <0.001 |
| 4 h Body Temp (˚C) | 35.8±0.1 (n = 6) | 32.9±0.1 (n = 5) | <0.001 |
| 24 h Body Temp (˚C) | 36.6±0.2 (n = 7) | 34.8±0.4 (n = 10) | 0.03 |
| Baseline HR (bpm) | 472±19 (n = 13) | 488±11 (n = 19) | 0.45 |
| Baseline EF (%) | 59.8±1.5 (n = 14) | 59.9±1.0 (n = 19) | 0.94 |
| 1 d HR (bpm) | 521±18 (n = 14) | 459±12 (n = 21) | 0.007 |
| 1 d EF (%) | 59.6±1.7 (n = 14) | 39.9±3.0 (n = 21) | <0.001 |
| 1 wk EF (%) | 59.6±2.3 (n = 5) | 41.4±3.4 (n = 6) | 0.002 |
| 4 wk EF (%) | 59.5±2.6 (n = 5) | 49.8±5.3 (n = 6) | 0.4 |

immediately after extubation. Arrest mice required an average of 1.32 minutes of CPR to achieve ROSC and were extubated after an average of 22.7 minutes (Table 1). The distribution of males and females is similar between groups (sham: 10 female, 9 male; arrest: 14 female, 16 male). While this study was not powered to examine sex-based changes amongst groups, surgical survival was not biased by sex distribution (S1 Table). There was no significant change to 1-day EF, CPR duration, or time to extubation between male and female arrest mice (S1 Table).

## SCA mice have temperature and HR dysregulation after ROSC

There were no significant differences in body temperature between groups at baseline (sham: 35.7±0.2˚C; arrest: 35.5±0.2˚C) (Table 1, Fig 1). Following extubation, mice were kept in a warmed recovery cage, and no difference was noted at 1 hour (sham: 35.9±0.1˚C; arrest: 35.9 ±0.3˚C) or 2 hours (sham: 35.8±0.1˚C; arrest: 35.2±0.3˚C). Arrest mice had significantly lower body temperatures once removed from the warming cage at 3 hours (sham: 35.8±0.1˚C; arrest: 33.0±0.6˚C, p<0.001), 4 hours (sham: 35.8±0.1˚C; arrest: 32.9±0.1˚C, p<0.001) and 24 hours (sham: 36.6±0.2˚C; arrest: 34.8±0.4˚C, p = 0.03). Arrest mice also had significantly lower HR one-day after SCA (sham 521±18 bpm; arrest 459±12 bpm, p = 0.007).

## SCA mice have increased 30-day mortality

A cohort of post-operative mice from each group was designated for survival studies over a 4-week time course. At 24 hours, 100% of sham mice survived (5 of 5) compared to 92% of arrest mice (12 of 13, p = 0.54). At 72 hours, 100% of sham mice were alive (5 of 5) compared to 46% of arrest mice (6 of 13, p = 0.052 vs sham). At 4 weeks, 100% of sham mice survived (5 of 5, median survival of 28 days) compared to only 38% of arrest mice (5 of 13, median survival 3 days, p = 0.03; Fig 1).

## SCA mice have reduced EF, which improves over time

Sham and arrest mice showed no difference in baseline EF (sham: 59.8±1.5%; arrest: 59.9 ±1.0%). One day after arrest, there was a significantly depressed EF in the arrest group (sham: 59.6±1.7%; arrest: 39.9±3.0%, p<0.001; Fig 2). EF of arrest mice remained significantly depressed 1 week after SCA procedure (sham: 59.6±2.3%; arrest: 41.4±3.4%, p = 0.002). Four weeks after arrest, there is no significance between EF of sham and SCA groups (Fig 2).

## SCA mice have evidence of prolonged ischemia after SCA and kidney damage at one day

As kidney damage is a common side effect of cardiac injury [13], we evaluated the duration of renal ischemia following SCA. A cohort of arrest mice (n = 10) were evaluated for kidney reperfusion following ROSC by evaluating renal artery flow. The mean duration of kidney ischemia was 20.6 minutes, with initial measurable kidney blood flow occurring on average 11.3 minutes after ROSC (Fig 3). Serum creatinine was significantly elevated in arrest mice at 1 day when compared to sham (sham: 0.36±0.06 mg/dL; arrest: 1.52±0.22 mg/dL, p<0.001; Fig 3), as was serum BUN (sham: 21.5±9.9 mg/dL; arrest: 156.0±39.8 mg/dL, p = 0.005). Semi-quantitative scoring of tubular injury was performed at the outer medulla and was noted to be higher in arrest mice (sham: 0.15±0.05; arrest: 3.33±0.29, p<0.0001).

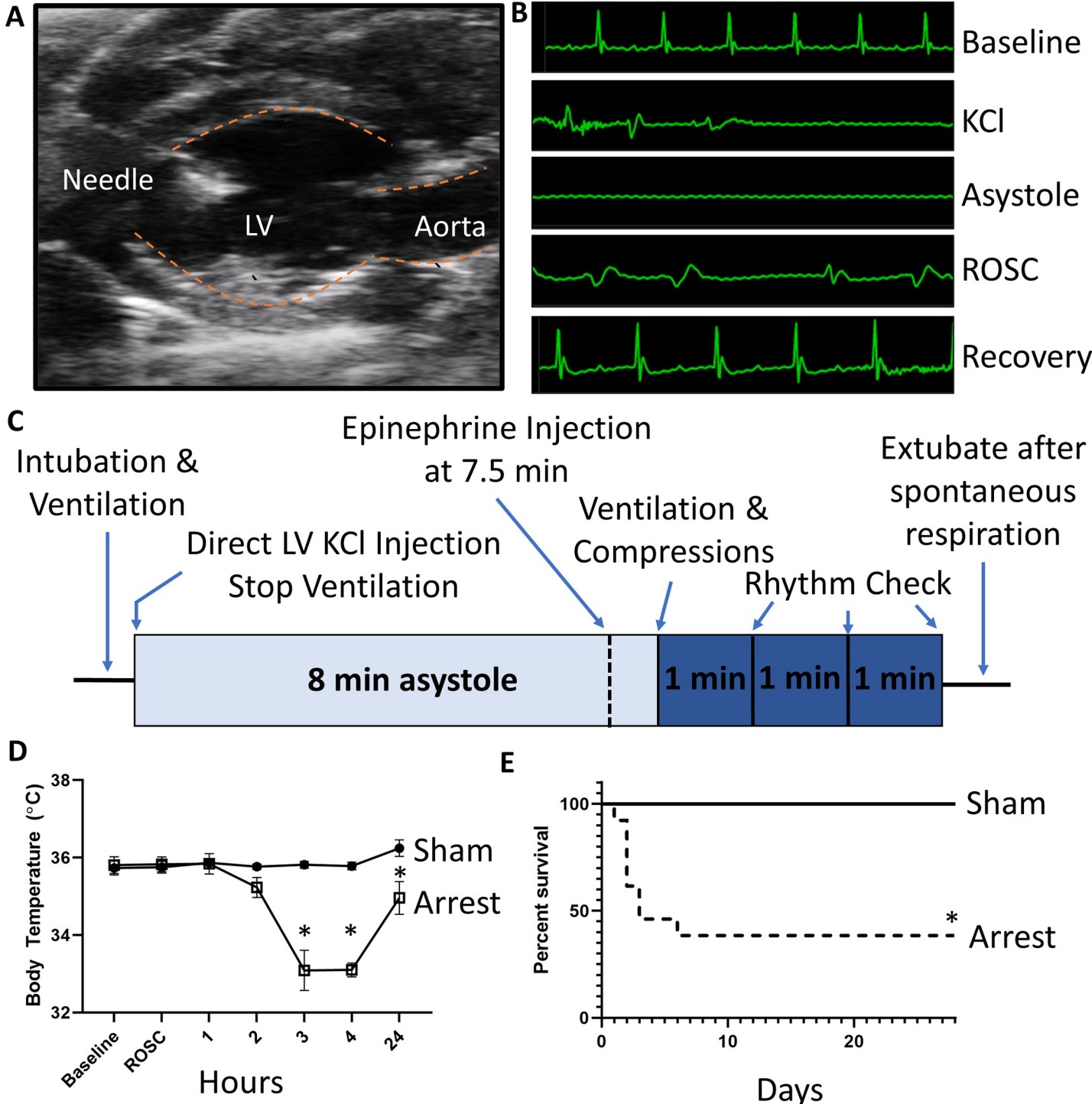

**Fig 1. Direct LV injection model of SCA.** A) Representative long-axis ultrasound image depicting introduction of a needle into the LV chamber. B) Representative ECG tracings at baseline, during KCl injection, during asystole, immediately after ROSC is achieved, and during recovery. C) Depiction of time course of SCA in this model. D) Temperature monitoring in sham and arrest mice at baseline, time of ROSC, and at 1, 2, 3, 4 and 24 hours post ROSC. Arrest mice have significantly depressed body temperature at 3, 4, and 24 hours after arrest when compared to sham mice. E) Mortality curve demonstrating descreased survival in arrest mice as compared to sham (initial sham n = 5; arrest n = 13). Data are expressed as mean +/- SEM. P-value: * < 0.05.

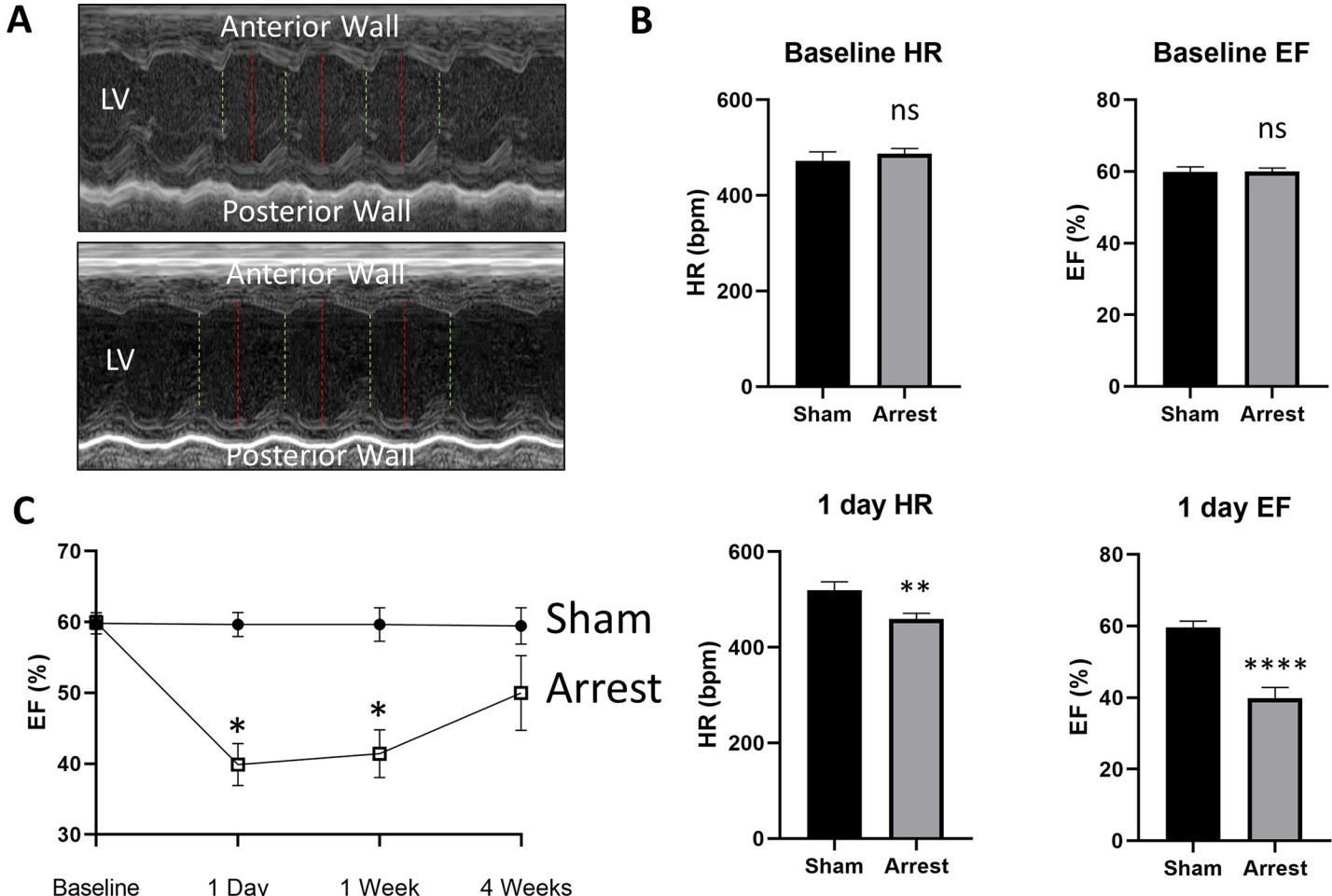

**Fig 2. Ejection fraction and heart rate in Sham and SCA Mice.** A) Representative M-mode tracings of a sham (top) and arrest (bottom) mouse one day after SCA, where red lines denote LV width during diastole and green lines denote systole. B) Heart rate (HR) and ejection fraction (EF) are similar between groups at baseline, but significantly depressed in SCA mice compared to sham at 1 day after arrest. C) EF is significantly decreased in SCA mice compared to sham at matched timepoints at 1 day and 1 week, but there is no significant EF change by 4 weeks. Data are expressed as mean +/- SEM. P-values: * < 0.05, ** < 0.01, *** < 0.001.

## SCA mice have diffuse tissue injury at one day

To assess systemic damage, additional serum assays were performed at 1 day to evaluate for liver damage (ALT), muscle damage (CK), tissue ischemia (lactate), and neurologic function. These assays were notable for a significant increase in ALT (sham: 47.6±4.7 U/L; arrest: 135.6 ±37.3 U/L, p = 0.047; Fig 3) in arrest mice compared to sham. No significant changes were noted to serum CK (sham: 1244±252 U/L; arrest: 1811±570 U/L, p = 0.36) or normalized serum lactate (sham: 1.00±0.06; arrest: 1.90±0.64, p = 0.15). Brief neurologic testing was performed as previously reported [14] on 14 sham and 17 arrest mice one day after SCA (Table 2). Two of the arrest mice were noted to have hind-leg ataxia and one mouse had sluggish movement; however, there was no significant difference in neurologic testing between the groups (sham: score 12.0±0; arrest: score 11.8±0.1, p = 0.13; Fig 3).

## Discussion

In this study, we modified the mouse model of SCA described by Hutchens et al. [9] by delivering KCl directly into the LV cavity under ultrasound guidance rather than IJ cannulation. This

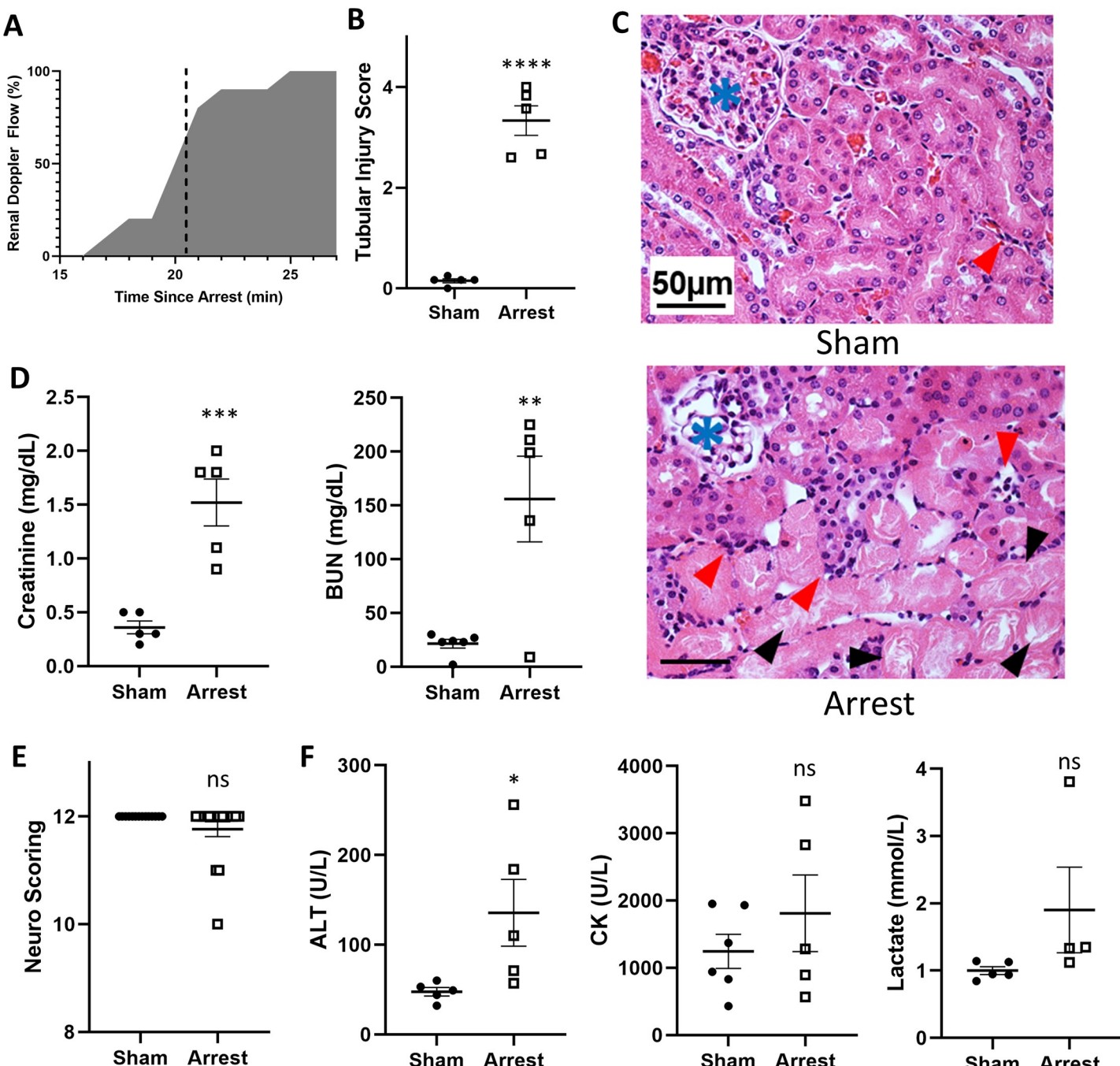

**Fig 3. Kidney, neurologic, and serum chemistry 1-day after SCA.** A) Percentage of mice with recovered kidney perfusion over time since arrest (n = 10). Mean recovery time was 20.55±0.68 min. B) SCA mice have increased kidney damage by semi-quantitative scoring of kidney injury in the outer medulla (n = 5/group). C) Representative H&E stains of sham (top) and arrest (bottom) mice one-day after arrest demonstrating proteinaceous casts in renal tubules (black arrowhead) and infiltrates (red arrowheads) with glomeruli marked (blue asterisk). D) Elevated serum creatinine and BUN in SCA mice at one-day. E) Neurologic scoring at one day. F) ALT, CK, and lactate changes at one day. Data are expressed as mean +/- SEM. P-values: *< 0.05, **< 0.01, ***< 0.001, ****< 0.0001.

delivery method causes immediate onset of asystole after KCl delivery in a highly controlled, easily visualized, and easily adoptable manner. Pre-clinical models of SCA are rarely performed in the mouse despite a large number of advantages of the murine model, including

**Table 2. 12-point neurologic function assessment for one-day Sham and SCA Mice.**

| Neurological Function | |
|---|---|
| **Level of Consciousness** | |
| No Tail Pinch Reflex | 0 |
| Weak Tail Pinch Reflex | 1 |
| Normal Tail Pinch Reflex | 2 |
| **Corneal Reflex** | |
| No Blink | 0 |
| Delayed Blink | 1 |
| Normal Blink | 2 |
| **Respiration** | |
| Irregular | 0 |
| Decreased Frequency with Normal Pattern | 1 |
| Normal Frequency and Pattern | 2 |
| **Righting Reflex** | |
| No Righting | 0 |
| Sluggish Righting | 1 |
| Rapid Righting | 2 |
| **Coordination** | |
| No Movement | 0 |
| Some Ataxia | 1 |
| Normal Coordination | 2 |
| **Activity** | |
| No Spontaneous Movement | 0 |
| Sluggish Movement | 1 |
| Normal Movement | 2 |
| **Total Possible Score** | **12** |

rapid development, low maintenance cost, and the opportunity for genetic manipulation [6, 7]. The low utilization of the murine model is likely attributable to surgical difficulties related to animal size. Groups that have embraced the mouse model of SCA almost uniformly rely on intravenous delivery of KCl for induction of cardiac arrest, but have utilized various durations of arrest (typically 4–16 minutes). These reports all use venous access either through the jugular or femoral veins for drug delivery. Establishing reliable venous access in the mouse remains a major barrier to wide-spread adoption of the mouse model of SCA.

Our model spares the use of major vessels for drug delivery by using direct LV introduction under ultrasound, resulting in procedure times around 30 minutes from anesthesia induction to extubation, with low surgical mortality (16.7% of SCA mouse surgical mortality, Table 1). While data is limited concerning procedure times in the traditional model, Abella et al. describe 50 minutes of anesthesia induction and up to 40 min of venous instrumentation prior to induction of asystole, followed by 2 hours of invasive monitoring [14]. More recently, Liu et al. report venous instrumentation followed by 15 min of stabilization, followed by asystole and CPR and then 30 min of invasive monitoring [15].

Ultrasound-guided catheter placement has already become a staple of hospital care for many clinicians, allowing physicians and researchers to easily transfer a known skill set into a translational model of SCA [16, 17]. As ultrasound continues to become more affordable and accessible [18], we anticipate continued translation of ultrasound approaches into preclinical models. Ultrasound utilization allows for continual non-invasive monitoring of cardiac

function throughout the arrest, resulting in precise assessment of ROSC and duration of asystole by intermittent checks, similar to human resuscitation efforts. Previously published models that utilize only ECG as an indicator of ROSC may falsely register pulseless electrical activity as a return of circulation, which may misidentify the time of asystole. Other models utilize an LV pressure catheter to accurately record restoration of cardiac flow; however, this requires the placement of an additional invasive canula. In the current report, the rapid procedure time, high survival, reduced surgical skill required, and venous sparing by this technique are highly advantageous over the traditional model.

Of the 30 mice that underwent arrest in this study, 25 survived the arrest and achieved ROSC, resulting in a relatively low mortality rate (16.7%) for the procedure (Table 1). This is a modest improvement over the 20% surgical mortality in the venous KCl mouse model described by Hutchens et al. [9] and the 27% mortality in a ventricular fibrillation mouse model described by Chen et al [8]. Only 1 of the 19 sham mice died during the procedure, which is likely attributable to surgical error. A subset of mice was studied for up to 4 weeks to assess long-term survival. At 72 hours, 6 of 13 arrest mice (46%) survived, which is in line with comparable recent studies publishing between 10 and 45% survival at 72 hours [19–21]. 5 of 13 mice (38%) survived for the entire duration of the study (Fig 1). The cause of death is likely multifactorial given the evidence of cardiac, renal, and liver damage. Previous works have attributed deaths to shock and neurologic injury in mouse models of SCA, though we are unable to verify these claims [14, 22]. We did not observe any evidence of pericardial effusion on echocardiography or free blood in the thorax suggestive of myocardial rupture in any of our mice.

After one day, EF in SCA mice is significantly decreased from 59.9% at baseline to 39.9%. These values are in-line with previously published one day EFs and are likely attributable to cardiac stunning [20]. EF remains significantly decreased at one week (41.4%), with improvement at four weeks (49.8%, Fig 2). These values and their relative improvement are similar to those observed in humans following cardiac arrest in the absence of coronary disease, as evidenced by a case series of cardiac arrest survivors, which noted 1 day EFs of 38%, 1 week EFs of 44%, and 2–3 week EFs of 50% [23].

Maintenance of body temperature is critically important to neurologic outcomes following SCA [24–26]. Baseline temperature was mildly depressed in both sham (35.7°C) and SCA mice (35.5°C), likely as a consequence of isoflurane anesthesia [27]. Following extubation, body temperature was maintained with active heating in a recovery cage for 2 hours. However, body temperatures fell significantly after active heating was stopped, which is consistent with post-arrest changes in humans [28] as well as previous mouse models of SCA [15, 29]. The SCA mice continued to show significant temperature dysregulation and depressed HR at one day (Fig 1, Table 1). We were unable to demonstrate significant neurologic deficit 24 hours after arrest by utilizing a well-established, 12-point examination, though some mice did display mild neurologic dysfunction [14]. Delayed, spontaneous hypothermia is known to be neuroprotective in other rodent cardiac arrest models and may explain the paucity of neurological injury noted [30]. We only noted ataxia in two mice and lethargy in one mouse following SCA, with no observable deficits in sham mice. While some groups are able to demonstrate neurologic injury with as little as 6-minutes of cardiac arrest [31], others have required extended arrest time of 12–14 minutes to detect neurologic changes [22, 29]. The 8-min time point was chosen as it is a well-established time-point in this field [9, 15, 32], however the procedure could be modified to allow for prolonged arrest time to study neurologic insult.

Mouse SCA models have been utilized to as a clinically-relevant model of both acute kidney injury [33, 34] (one day after arrest) and chronic kidney disease (seven weeks after arrest, attributed to prolonged inflammation after reperfusion) [34]. We show that our model of SCA

similarly develops markers of AKI, as evidenced by elevated serum creatinine, BUN, and tubular damage 1 day after arrest (Fig 3). Kidney injury is not typically apparent with 8 minutes of direct ischemia, which typically require 15–20 minutes for the development of focal injury [35]. By utilizing doppler ultrasonography of the renal artery, we found that the kidney did not receive measurable perfusion until 11.25 minutes after ROSC, for a total mean ischemia duration of 20.55 minutes, resulting in kidney ischemia times consistent with established direct ischemia reperfusion injury models (Fig 3) [36, 37]. Previous work by Ikeda et. al. reports partial restoration of cortical blood flow within minutes of ROSC which gradually improves over the next twenty minutes [38], which is in line with our estimates of restoration of renal perfusion by renal artery flow [38].

Finally, our model suggests the presence of global ischemic injury following SCA. ALT, a non-specific marker of liver injury, is significantly elevated one-day after SCA (Fig 3). Some mice had elevated levels of CK, a non-specific marker of muscle degradation, and serum lactate, but these changes were not consistent in the SCA group. While these changes are suggestive of secondary ischemic damage as seen in post-cardiac arrest patients, larger experiments would be necessary to fully characterize these changes in our model. We anticipate that these changes could become significant with prolonged arrest duration, but at the cost of increased mortality. Further evaluation of this multiple organ damage and the degree to which each organ system is involved may be an important step toward improving recovery and guiding post-cardiac arrest interventions.

## Limitations

While the ultrasound-guided model of SCA offers advantages over the currently established intravenous model, there are a number of limitations to its use. First, the operator was not blinded to sham vs SCA groups in our study, as the sham groups only received a single injection of epinephrine rather than an injection of KCl followed by an injection of epinephrine in the arrest mouse. Second, there is potential myocardial injury related to the apical puncture by needle, however, we did not observe any pericardial effusion or blood in the thorax suggestive of myocardial rupture. Additionally, KCl is an artificial method for inducing asystole that may not fully recapitulate arrhythmic models of SCA, which may have residual coronary flow and intermittent LV contraction.

## Conclusions

We demonstrate a novel mouse model of SCA that utilizes direct LV injection of KCl under ultrasound guidance that allows for rapid and reliable arrest with low surgical mortality. This model develops significant cardiac, kidney, and liver injury at one day as well as neurologic injury in some animals. This model lowers the barrier to entry for establishing a mouse model of SCA, which will help researchers investigate the mechanisms underlying SCA mortality.

## Supporting information

**S1 Table. Physiologic and surgical characteristics between sexes.**
(DOCX)

## Author Contributions

**Conceptualization:** Cody A. Rutledge, Cameron Dezfulian, Brett A. Kaufman.

**Data curation:** Cody A. Rutledge.

**Formal analysis:** Cody A. Rutledge, Takuto Chiba, Sunder Sims-Lucas.

**Funding acquisition:** Brett A. Kaufman.

**Investigation:** Cody A. Rutledge, Takuto Chiba, Kevin Redding.

**Methodology:** Cody A. Rutledge.

**Resources:** Cameron Dezfulian, Brett A. Kaufman.

**Supervision:** Cameron Dezfulian, Sunder Sims-Lucas, Brett A. Kaufman.

**Visualization:** Cody A. Rutledge, Takuto Chiba.

**Writing – original draft:** Cody A. Rutledge, Cameron Dezfulian, Sunder Sims-Lucas.

**Writing – review & editing:** Cody A. Rutledge, Cameron Dezfulian, Brett A. Kaufman.

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
