## [Decision Letter · Decision Letter 0]

26 Aug 2020

PONE-D-20-22662

A novel ultrasound-guided mouse model of sudden cardiac arrest

PLOS ONE

Dear Dr. Kaufman,

Thank you for submitting your manuscript to PLOS ONE. After careful consideration, we feel that it has merit but does not fully meet PLOS ONE’s publication criteria as it currently stands. Therefore, we invite you to submit a revised version of the manuscript that addresses the points raised during the review process.

Two out of three reviewers discussed the added utility of the model compared to previously described models, therefore the authors should ensure the benefits of this models are well described. Furthermore, the authors should address all of the comments of the reviewers.

We look forward to receiving your revised manuscript.

Kind regards,

Daniel M. Johnson, PhD

Academic Editor

PLOS ONE

Journal Requirements:

2. Please address the following:

- Please ensure you have thoroughly discussed any potential limitations of this study within the Discussion section.

- In your Methods section, please ensure you have described the source of the mice used in this study.

Reviewers' comments:

Reviewer's Responses to Questions

**Comments to the Author**

1. Is the manuscript technically sound, and do the data support the conclusions?

Reviewer #1: No

Reviewer #2: Yes

Reviewer #3: Partly

2. Has the statistical analysis been performed appropriately and rigorously? 

Reviewer #1: Yes

Reviewer #2: Yes

Reviewer #3: Yes

3. Have the authors made all data underlying the findings in their manuscript fully available?

Reviewer #1: No

Reviewer #2: Yes

Reviewer #3: No

4. Is the manuscript presented in an intelligible fashion and written in standard English?

Reviewer #1: Yes

Reviewer #2: Yes

Reviewer #3: Yes

5. Review Comments to the Author

Reviewer #1: This is a very interesting manuscript which details creation and validation of a mouse model of sudden cardiac arrest. While previous models have been limited by requirement of vessel cutdowns which increase procedure times, the authors use direct cardiac puncture to administer KCl for induction of asystole. They report outcomes, including survival and end-organ damage in both a SCA and sham group. Overall, I found this manuscript well written, the topic novel, and the model of potential value to the literature. There are several lingering questions which I believe need to be addressed to strengthen the validity of this model.

It is surprising that asystole secondary to KCl administration, which disrupts the resting membrane potential of the cardiac myocyte could be overcome with just epi. I would have thought that reduction in hyperkalemia would be needed. What was the KCl level in the blood post arrest after ROSC and what was it in comparison to when asystole was achieved? Was serum K truly normal (or close to normal) during ROSC?

More detail is needed regarding the sham mice and the sham procedure. Was the saline vehicle injected? Were the mice given sham CPR (does not appear so)?

Was there any arrhythmia noted with ventricular puncture or administration of either the sham or KCl?

What was the cause of death of the one sham mouse that died? Asystole? Did other mice have arrhythmias or had asystole then underwent ROSC?

The authors do not clearly state that they were blinded during the administration of the KCl vs the sham injection. This is a critical point which needs to be done to prevent bias. Particularly given that at least one sham mouse died and potentially more had arrhythmias or short-lived asystole?

There are several parts of the methods which require more detail: 1) Prior to injection, where the reagents warmed to 37C or kept at room temperature, 2) how was the ECG performed, 3) how was the intubation performed and the mouse ventilated, 4) how was CPR performed

Epi was administrated at the same dose regardless of the size of the mice. As there can be significant differences in the weight of mice of different sexes, why dosing was not adjusted based on weight should be justified.

Overall, the body temperature of the mice was lower than anticipated for both sham and SCA. It was even lower for the SCA mice. Why is this? Is it a sign of low cardiac output? Hemodynamic instability?

In Table 1, it would appear that 25 out of 30 mice in the SCA group survived. In the section on 30 day mortality, there are only 13 mice at 24 hours? Did the difference (12 mice) die after ROSC but before the 24 hour time point? In the sham, it says only 5 mice were present at 24 hours but 18 survived the procedure?

Putting this model forward as a model for SCA is critical for understanding neurologic outcomes. It would be most interesting to see brain histology of the mice which had reduced neuro scoring.

Reviewer #2: The study by Cody A Rutledge al reports on the use of a new mouse model of cardiac arrest.

First of all the reviewer would like to acknowledge the work by the authors by advancing the field of experimental cardiac arrest, however some concerns exist.

I must admit I disagree with the primary premise for the development of this model. The authors write “limited by challenges with surgical technique and reliable venous access”

Is this really true. I don’t know how many papers that have been published using reliable venous access in the mouse. Instead, researchers now have to purchase extremely expensive ultrasound equipment?

Secondly, in my opinion, we don’t need more simple models in the field of cardiac arrest, we need more clinical relevant models of cardiac arrest. The model described in this paper is not an example of that.

For example the authors look at mortality over time. With this model we have no idea what the cause of death is. Respiratory, neurological, cardiac, renal? This is very important when looking at treatment effects.

Is potassium induced cardiac arrest truly clinical relevant?

The authors also write “life support related to the animal’s size7,8.” This is still true.

The authors write “vessel post-operatively, which may further contribute to organ damage and alter outcomes.” Yet the same authors use rat models of cardiac arrest where they due tie of vessels post-operatively?

Page 4 line 77 “If a ventricular rhythm” what do the authors actually mean? A ventricular rhythm is VT/VF or do the authors mean a possible pulse generating rhythm or a supra-ventricular rhythm? ‘

The authors euthanize the animals if they have not received ROSC at 3 minutes? Why ? Again this not very clinical relevant?

Why 7.5 minutes of arrest? please justify this in the paper.

Why did the authors focus on kidney function with both blood samples and histology? Why not neurological injury that is the major driver of death?

The authors are acknowledged for using both males and females.

Since hypothermia in animal models is highly protective it makes no sense to remove animal from the heating chamber and allowing them to develop hypothermia

Page 7 section SCA mice have increased 30-day mortality

This is a long section and a lot of text that clearly demonstrated in figure 1. Is this really necessary.

Page 8 line 196 “there was no significant difference in neurologic testing”

Why develop a cardiac arrest model without neurological injury? This is a major limitation of the model.

Please revive figure 1. They authors don’t perform a pulse check. Pulse check is a clinical feature the authors look for low. This is not the same.

Reviewer #3: The authors address an important problem, ischemia-reperfusion injury, with enhancements to a well-characterized mouse model, cardiac arrest and cardiopulmonary resuscitation (CA/CPR). The difficulty of the established model limits use and replication of results, perhaps impairing the vertical advance of the science. The authors describe several critical enhancements to the model, which it is suggested increase the utility and perhaps generalizability of findings. Improvement in this model, which is critical to investigations in several disciplines, could indeed substantially advance the science of ischemia-reperfusion injury, and consequently have impactful results in the care of patients with critical illness. Therefore this manuscript is appropriate for the journal, and has potential for significant impact. However, some significant concerns should be addressed prior to consideration for publication.

Major concerns:

The major claim of the manuscript is that use of ultrasound to directly deliver potassium to the LV increases access to the model because it reduces the barrier imposed by the difficulty of obtaining IV access in mice. There are two problems with this claim: 1)There is no data to support the claim that ultrasound-guided injection is more easily performed, or more accessible to other labs than IV access followed by IV injection of potassium, and 2)it is at least a little implausible that cardiac ultrasound in the mouse is easier or more accessible to many labs than IV catheter placement given the cost of ultrasound machines and the expertise required to use them in mice. If the authors want to make this claim, they need to provide some data. There are important advantages to this model, and to the technique, but accessibility is unlikely to be the most important one. Therefore, it is suggested that the authors confine the claims made in the conclusion to specific and verifiable claims regarding time, effort, survival, comparability of injury, and time to train, rather than vague claims of accessibility. Other groups have reported the time to prepare a mouse for CA/CPR. This data, if relevant, should be included.

The second claim is that preservation of vessels used for cannulation in other versions of the model reduces procedural times and prevents complications related to these vessels being cannulated or blocked. No evidence is presented that in other models, vessels are blocked, although it’s certainly plausible. Without this evidence, this claim is not verifiable, and like the first claim, is not very important. Similarly, the authors ignore the injury unique to their model, which is 2 30-gauge holes in the left ventricle. Some mention of this, and possible effects, should be made. Were the hearts assessed for injection-related injury at necropsy?

It is not appropriate to report “a trend toward neurologic injury” as a conclusion, as the statistical analysis demonstrates no difference from control. “neurologic injury in some animals” would be an appropriate conclusion. Other authors have reported difficulty obtaining significant neuronal injury at CA times consistent with organism survival, and at least one group has resorted to heating the head while cooling the body (see, eg, Noppens et al, PMID 18957991) to increase neuronal injury while reducing systemic injury. This history would provide context for the finding of minimal neurologic injury in the present model.

Really, the most important result of this enhancement is the use of intermittent compressions with pulse checks, similar to human CPR. The authors should comment on this.

Minor concerns:

The authors used mice of both sexes, but provide no data on sex differences, which might be important in the overall results. Please provide differential analysis based on sex, or at least denote data points contributed by males vs females in figures.

Description of temperature dysregulation after CA in mice is not novel, and should be discussed in concert with relevant literature.

How was temperature monitored on an hourly basis in awake mice?

Were mice reanesthetized for renal ultrasound after ROSC? Were the results from these mice, which may have been affected by additional isoflurane (which has protective effects in ischemia-reperfusion injury) analyzed in the same groups as mice without renal ultrasound?

Authors describe this as a model of “sudden cardiac arrest” (SCA), but SCA is really a disease of ventricular tachycardia leading to asystole rather than a disease of asystole due to inability to conduct an impulse (from potassium). Authors should justify the use of the SCA term.

The authors describe anucleated proximal tubular epithelial cells as a sign of nephrotic damage. This is not the correct meaning of the term “nephrotic”, which refers to proteinuria.

The authors report there was no renal blood flow by Doppler until 20 minutes post-ROSC. This is not consistent with the data reported by Hutchens et al (same group as reference 9, PMID 20068453). Reasons for the difference, which would seem important, as the Hutchens paper reports flow measured in the cortex within 5 minutes of ROSC, while the current manuscript reports no flow in the renal artery at that time, would seem important to discuss. It would also be relevant to include discuss relevant or comparable data obtained from humans, if this is available.

6. PLOS authors have the option to publish the peer review history of their article (what does this mean?). If published, this will include your full peer review and any attached files.

Reviewer #1: **Yes: **Andrew Landstrom

Reviewer #2: No

Reviewer #3: No

---

## [Author Response · Author response to Decision Letter 0]

23 Sep 2020

Please see uploaded file with colored text for easier reading.

Response to reviewer comments:

Reviewer #1: This is a very interesting manuscript which details creation and validation of a mouse model of sudden cardiac arrest. While previous models have been limited by requirement of vessel cutdowns which increase procedure times, the authors use direct cardiac puncture to administer KCl for induction of asystole. They report outcomes, including survival and end-organ damage in both a SCA and sham group. Overall, I found this manuscript well written, the topic novel, and the model of potential value to the literature. There are several lingering questions which I believe need to be addressed to strengthen the validity of this model.

We appreciate Reviewer #1’s recognition of the value of this manuscript to the field, as well as the thoughtful and reasonable critiques provided. We address the individual comments below.

It is surprising that asystole secondary to KCl administration, which disrupts the resting membrane potential of the cardiac myocyte could be overcome with just epi. I would have thought that reduction in hyperkalemia would be needed. What was the KCl level in the blood post arrest after ROSC and what was it in comparison to when asystole was achieved? Was serum K truly normal (or close to normal) during ROSC?

Previous works have reported the evaluated serum potassium levels by KCl-induced cardiac arrest in the mouse. Abella et al. (Circulation, 2004) reported a blood potassium level of 10.6±0.4 mEq/L during arrest that returned to normal levels within 60 min of ROSC. Kofler et al. (Journal of Neuroscience Methods, 2004) reported potassium levels of 8.0 mEq/L after 5 min of arrest that return to normal within 15 min. Both of these works are cited in our manuscript. We believe the crucial factor for resuscitation is removing potassium from the coronary arteries, which we achieve by injecting a large volume of fluid into the LV after asystole. Anecdotally, saline alone will work to clear the KCl from the heart; however, we use epinephrine in saline for the second injection as it is clinically relevant and in line with previous mouse models.

More detail is needed regarding the sham mice and the sham procedure. Was the saline vehicle injected? Were the mice given sham CPR (does not appear so)?

We appreciate the attention to detail regarding our methods and have updated the methods to include additional details regarding the sham surgery. The cardiac arrest mice underwent an injection of 40 uL of KCL followed 8 min later by a second injection of 500 uL of epinephrine. In comparison, the sham mice were only treated with a single LV injection of 500 uL of epinephrine. The sham mice were not treated with an initial vehicle injection. The sham mice did not receive any CPR. 

We realized that the arrest mice were subjected to two LV punctures, but for the second injection, the heart was not moving. The sham mice only received one puncture to mirror the single injection into a beating heart of the arrest group

We have updated the following sections of our manuscript to clarify these points:

Methods, paragraph 2: “Sham mice received no KCl injection. Rather, sham animals were treated with a single direct LV injection of 500 µL epinephrine in normal saline. Sham animals did not experience asystole or receive chest compressions, and were extubated within minutes after injection.”

Additionally, we have added a limitations section to the manuscript for further clarification:

Limitations, paragraph 1: “While the ultrasound-guided model of SCA offers advantages over the currently established intravenous model, there are some limitations to its use. First, the operator was not blinded to sham vs. SCA groups in our study, as the sham groups only received a single injection of epinephrine rather than an injection of KCl, followed by an injection of epinephrine in the arrest mouse…”

Was there any arrhythmia noted with ventricular puncture or administration of either the sham or KCl?

Premature ventricular contractions, typically 1-3 beats, can be noted on ECG during the needle puncture, but no sustained arrhythmias were recorded.

This detail has been added to Methods, paragraph 2: “Occasional premature ventricular contractions were noted during needle puncture, typically 1-3 beats, but no sustained arrhythmias were recorded.”

What was the cause of death of the one sham mouse that died? Asystole? Did other mice have arrhythmias or had asystole then underwent ROSC?

We cannot state the cause of death conclusively in the sham mouse, and for this reason, have not included it in the manuscript. The mouse died immediately after extubation and was no longer being monitored by ECG. We suspect the death was related to surgical airway and potentially airway trauma, though we cannot be certain. 

Rarely, short runs of ventricular tachycardia were noted on ECG following ROSC in arrest mice, but all of these episodes self-terminated after a few seconds. Additional details are provided in the manuscript reporting that 4 of the 5 SCA mice that did not survive the surgery were related to lack of ROSC, whereas the final SCA mice that did not survive the surgery died immediately following extubation.

We have updated Results, paragraph 1: “A single mouse from the sham group (1/19, 5.2%) died immediately following extubation and five mice from the arrest group (5/30, 16.7%), where four did not achieve ROSC and one died immediately after extubation.”

The authors do not clearly state that they were blinded during the administration of the KCl vs the sham injection. This is a critical point which needs to be done to prevent bias. Particularly given that at least one sham mouse died and potentially more had arrhythmias or short-lived asystole?

As detailed in the comments above, the operator was not blinded to groups as the sham mice only received a single injection. There is no simple method for blinding in this scenario, as the arrest mice undergo immediate asystole, whereas the sham mice do not. Histologic analysis, serum analysis, and echocardiography were all measured in a blinded fashion.

We have added to Methods, paragraph 4: “Echocardiography was performed at baseline, 1 day, 1 week, and 4 weeks as previously described10 and measured by a blinded operator. 

Methods, paragraph 6 previously indicated: “Histological scoring was performed in a blinded fashion at 40x magnification on outer medullary regions of the tissue sections…”

There are several parts of the methods which require more detail: 1) Prior to injection, where the reagents warmed to 37C or kept at room temperature, 2) how was the ECG performed, 3) how was the intubation performed and the mouse ventilated, 4) how was CPR performed

We again appreciate the reviewer’s attention to detail and have amended our methods section to include changes. Reagents were warmed under a heat lamp to 37 °C before injection. ECG was monitored via lead 2 of a surface ECG system integrated into the Visual Sonics echocardiography system. Intubation was performed by quickly placing a 22-g catheter endotracheally in the supine position while the animal was anesthetized. CPR was performed as detailed by Hutchens at al. (JOVE, 2011), including a rate of 300 bpm, compressions aimed to a depth of 1/3 of 1/2 of the thorax depth, and pressure placed just laterally of the xiphoid process. These changes are noted below:

Methods, paragraph 1 now reads: “mice were placed in a supine position and quickly intubated while anesthetized by placing a 22 g catheter endotracheally and then mechanically ventilated…”

Methods, paragraph 2: “40 µL of 0.5M KCl in saline (pre-warmed to 37°C) was delivered into the LV cavity, causing immediate asystole as observed on lead 2 of surface ECG…”

Methods, paragraph 2: “At 8 minutes, the ventilation was resumed at 180 bpm and CPR initiated. CPR was performed manually by finger compressions just lateral of the xiphoid process, aiming for a depth of 1/3 to 1/2 of the animal’s thorax, at 300 bpm for 1 min.”

Epi was administrated at the same dose regardless of the size of the mice. As there can be significant differences in the weight of mice of different sexes, why dosing was not adjusted based on weight should be justified.

We followed methods previously published by Hutchens et al. (Jove, 2011), which utilize standard KCl dosing between male and female mice. We have included in our updated manuscript a new table, Supplemental Table 1, describing differences between sex the sham in arrest mice. While there is a significant difference in body weight between male and female mice, there was no change to CPR duration, time to extubation, or survival between male and female mice. Because we noted no differences in time to ROSC between groups, we felt it was appropriate to maintain stable dosing between all mice.

Overall, the body temperature of the mice was lower than anticipated for both sham and SCA. It was even lower for the SCA mice. Why is this? Is it a sign of low cardiac output? Hemodynamic instability?

Initial body temperatures for sham mice averaged 35.7 °C and for SCA mice 35.5 °C. While these values are below the 37 °C anticipated for a mouse, the initial values were taken while anesthetized with isoflurane, which is known to lower body temperatures (Tsukamoto et al., Experimental Animals, 2015). This reference has been added to our manuscript as noted below. The body temperature of the sham mice returned to 36.6 °C one day after the procedure. The SCA mice, however, had a dramatic temperature drop that is most noticeable after removing the mice from the heated recovery cage, as evidenced by temperature drops at 3 hours, 4 hours, and 24 hours. Dezfulian et al. similarly reported depressed body temperature at 24 hours, which they attributed to neurologic injury (Circulation, 2009). They found that 24-hour body temperature was improved by employing neuro-protection with nitrite therapy at the time of arrest. At the same time, there was no difference in blood pressure between their treatment groups. Their work suggests that neurologic injury rather than hemodynamic compromise drives hypothermia. While we reference this manuscript, supporting these claims is beyond the scope of this paper.

Discussion, paragraph 6: “Baseline temperature was mildly depressed in both sham (35.7 °C) and SCA mice (35.5 °C), likely as a consequence of isoflurane anesthesia27”

In Table 1, it would appear that 25 out of 30 mice in the SCA group survived. In the section on 30 day mortality, there are only 13 mice at 24 hours? Did the difference (12 mice) die after ROSC but before the 24 hour time point? In the sham, it says only 5 mice were present at 24 hours but 18 survived the procedure?

5 sham mice and 13 mice were designated into a 4-week survival cohort, which is reported in the mortality curve in Figure 1. Additionally, we provide surgical survival data for all the animals that underwent surgery (19 shams, 30 arrests) in table 1. To clarify this, we have added additional language to our methods section:

Methods, paragraph 3: “Survival Analysis. A cohort of sham and SCA mice that survived the initial surgery were designated for survival analysis over a 4-week time frame. 5 sham mice and 13 SCA mice were included in this study. Survival was assessed every morning over 4 weeks post-operatively.”

Putting this model forward as a model for SCA is critical for understanding neurologic outcomes. It would be most interesting to see brain histology of the mice which had reduced neuro scoring.

We performed preliminary brain histology on a small cohort of mice to evaluate for neuronal damage in the CA1 region of the hippocampus at one day. There was no clear evidence of neuronal death. This observation is consistent with the report by Dezfulian et al. (Circulation, 2009). In the Dezfulian manuscript, mice underwent 12-minutes of asystole compared to the 8-min in our model. Dezfulian et al. were not able to show histologic evidence of brain ischemia at 24 hours but did note significant changes by 72-hours after ROSC. We found no damage in pilot studies in our mice and did not pursue staining for a larger cohort of animals.

Reviewer #2: The study by Cody A Rutledge al reports on the use of a new mouse model of cardiac arrest.

First of all the reviewer would like to acknowledge the work by the authors by advancing the field of experimental cardiac arrest, however some concerns exist.

I must admit I disagree with the primary premise for the development of this model. The authors write “limited by challenges with surgical technique and reliable venous access”

Is this really true. I don’t know how many papers that have been published using reliable venous access in the mouse. Instead, researchers now have to purchase extremely expensive ultrasound equipment?

We appreciate Reviewer #2’s acknowledgment of advancing the field and have addressed each of the critiques. We feel that our model offers some advantages over the established model, which relies on establishing venous access. We have clarified this argument by including further details about the current venous procedure, though published data on procedure times and time to adopt the appropriate surgical technique is sparse. We have included in our introduction a reference to Hutchens et al. (Jove, 2011) that describes in detail the current venous procedure. In their manuscript, they estimate that 30-50 mice may be needed to establish a reliable surgical model. Additionally, we have added to our discussion published data concerning procedure times for the venous model:

Discussion, paragraph 2: “While data is limited concerning procedure times in the traditional model, Abella et al. describe 50 minutes of anesthesia induction and up to 40 min of venous instrumentation prior to induction of asystole, followed by 2 hours of invasive monitoring14. More recently, Liu et al. report venous instrumentation followed by 15 min of stabilization, followed by asystole and CPR and then 30 min of invasive monitoring15.”

Anecdotally, we found the traditional procedure too burdensome for adoption in our laboratory, which was the driving factor of this manuscript. We established this ultrasound-guided approach rapidly, including on the very first animal, and have found highly reliable results since that time.

Regarding ultrasound availability, we believe that ultrasound is increasing in prevalence and will gain favor as a preclinical surgical tool. We have added a reference to a relevant meta-analysis in our Discussion concerning the adoption of ultrasound into clinical practice. We have not identified a metanalysis for preclinical ultrasound utilization; however, it should be noted that Visual Sonics claims that 3900 publications have used their equipment in preclinical models. While it is difficult to provide references concerning the cost or availability of this equipment, we feel that ultrasound will continue to become more common and that adopting these techniques will aid laboratories in their animal studies. While we utilize the Vevo 3100 for published data, we have successfully performed this procedure with a Vevo 700 series as well as a Philips Lumify handheld vascular ultrasound probe (retail ~$7000).

The Discussion about clinical ultrasound adoption is included in Discussion, paragraph 3: “Ultrasound-guided catheter placement has already become a staple of hospital care for many clinicians, allowing physicians and researchers to easily transfer a known skill set into a translational model of SCA16,17. As ultrasound continues to become more affordable and accessible18, we anticipate continued translation of ultrasound reliance into preclinical models.”

Secondly, in my opinion, we don’t need more simple models in the field of cardiac arrest, we need more clinical relevant models of cardiac arrest. The model described in this paper is not an example of that.

In this manuscript, we provide evidence for a rapid and reliable mouse model of SCA. Our objective is not to provide a model with the highest clinical fidelity, but rather to improve the current mouse model. 

For example the authors look at mortality over time. With this model we have no idea what the cause of death is. Respiratory, neurological, cardiac, renal? This is very important when looking at treatment effects.

We have been unable to identify a uniform or definitive cause of death in our mice, as it is likely a multifactorial process. The original manuscripts describing a KCl model of SCA in the mice attribute (Abella et al., Circulation, 2004 and Kofler et al., Journal of Neuroscience Methods, 2004) speculate that “shock” is the driver of early deaths (within 6 hours) while neurologic injury contributes to later death. These appear plausible, but we are unable to confirm or refute these claims conclusively. We have included two sentences in the Discussion to this effect:

Discussion, paragraph 4: “The cause of death is likely multifactorial given the given the evidence of cardiac, renal, and liver damage. Previous works have attributed deaths to shock and neurologic injury in mouse models of SCA, though we are unable to verify these claims14,22. We did not observe any evidence of pericardial effusion on echocardiography or free blood in the thorax suggestive of myocardial rupture in any of our mice.”

Is potassium induced cardiac arrest truly clinical relevant?

We appreciate the reviewer’s concern regarding clinical relevance. Potassium induced cardiac arrest is an exceedingly rare clinical occurrence but remains the only reliable method for inducing arrest in the mouse. While some papers have been published with electrical or asphyxial means of SCA, these models are unreliable and difficult (See Abella, Circulation 2004 and Vognesen, Resuscitation 2017). We have included the use of KCl as a limitation to this paper:

Limitations, paragraph 1: “Additionally, KCl is an artificial method for inducing asystole that may not fully recapitulate arrhythmic models of SCA, which may have residual coronary flow and intermittent LV contraction.”

The authors also write “life support related to the animal’s size7,8.” This is still true.

The authors write “vessel post-operatively, which may further contribute to organ damage and alter outcomes.” Yet the same authors use rat models of cardiac arrest where they due tie of vessels post-operatively?

We appreciate that this reviewer has read our co-authors’ publications. Dr. Cameron Dezfulian has previously published with both mouse and rat models that utilize vessel delivery, necessitating the ligation of vessels as well as asphyxia and electrically induced models. We believe that our experience with multiple models of SCA and our knowledge of their limitations strengthens our argument for the adoption the ultrasound-guided model.

Page 4 line 77 “If a ventricular rhythm” what do the authors actually mean? A ventricular rhythm is VT/VF or do the authors mean a possible pulse generating rhythm or a supra-ventricular rhythm?

We apologize for this confusion. “Ventricular rhythm” is meant to signify and ECG signal generated by the ventricle, namely, a QRS complex. We have changed the term “ventricular rhythm” in paragraph 2 of the methods section to “sinus rhythm” for clarity. We did not observe any VF or sustained VT in our model.

The authors euthanize the animals if they have not received ROSC at 3 minutes? Why ? Again this not very clinical relevant?

The 3-minute time point was drawn from work by Hutchens et al. (JOVE, 2011). Kofler et al. (Journal of Neuroscience Methods, 2004) terminated CPR if no ROSC was obtained at 2.5 min while Abella et al. (Circulation, 2004) completed up to 5 min of compressions. In our experience, animals that did not regain ROSC by 3 minutes have no meaningful survival and were deemed surgical failures.

Why 7.5 minutes of arrest? please justify this in the paper.

We apologize for any confusion. Our mice had a total of 8 min of asystole prior to CPR. After receiving KCl, mice remained on the table for 7.5 min, followed by epinephrine injection over 30 seconds, totaling 8 min of asystole, at which time CPR was started. We have updated Figure 1 to reflect better the 8 min of asystole, which is further detailed in our methods section. 8 min is a common time point in mouse models of arrest as published by Ikeda et al. (Nitric Oxide, 2015), Hutchens et al. (JOVE, 2011), and Liu et al. (Aging and Disease, 2018). We have added a sentence in our Discussion clarifying this point:

Discussion, paragraph 6: “The 8-min time point was chosen as it is a well-established time-point in this field9,15,32; however, the procedure could be modified to allow for prolonged arrest time to study neurologic insult.”

Why did the authors focus on kidney function with both blood samples and histology? Why not neurological injury that is the major driver of death?

The focus of our laboratory is on cardiovascular (Kaufman) and renal disease (Sims-Lucas), which is why the endpoints in this paper focus on cardiac and kidney injury. We performed a brief neurologic examination as detailed in Table 2 and Figure 3, but were unable to demonstrate significant changes. As noted above in response to Reviewer #1, we did perform pilot studies for histologic assessment of damage to the hippocampus but were unable to demonstrate any changes. This finding is consistent with previous work, in which neurologic damage is not noticeable on histology until 3-days after ROSC (Dezfulian, Circulation 2009). We believe that the experiments described in this manuscript can be scaled to study neurologic injury, as described in our discussion section in paragraph 6.

The authors are acknowledged for using both males and females.

Since hypothermia in animal models is highly protective it makes no sense to remove animal from the heating chamber and allowing them to develop hypothermia

We describe in this manuscript a novel model for cardiac arrest and the resulting pathophysiology and do not venture into any interventions at improving these outcomes here. Animals were placed in a heated recovery cage for 2-hours after SCA, as is standard for mouse surgical models. We feel that manipulating body temperature to modify post-SCA outcomes is beyond the scope of this paper. We do, however, provide references to papers interested in therapeutic temperature management.

Page 7 section SCA mice have increased 30-day mortality

This is a long section and a lot of text that clearly demonstrated in figure 1. Is this really necessary.

At the reviewer’s suggestion, we have modified this paragraph for the sake of brevity. The updated paragraph reads: “A cohort from each group was designated for survival studies over a 4-week time course. At 24 hours, 100% of sham mice survived (5 of 5) compared to 92% of arrest mice (12 of 13, p=0.54). At 72 hours, 100% of sham mice were alive (5 of 5) compared to 46% of arrest mice (6 of 13, p=0.052 vs. sham). At 4 weeks, 100% of sham mice survived (5 of 5, median survival 28 days) compared to only 38% of arrest mice (5 of 13, median survival 3 days, p=0.03; Figure 1).”

Page 8 line 196 “there was no significant difference in neurologic testing”

Why develop a cardiac arrest model without neurological injury? This is a major limitation of the model.

Our primary research interests concern cardiovascular and renal outcomes and have chosen to evaluate SCA time-points that highlight injury to those systems. Both the time of arrest and time of animal monitoring can be extended to focus on neurologic outcomes, as described in the discussion section, paragraph 6.

Please revive figure 1. They authors don’t perform a pulse check. Pulse check is a clinical feature the authors look for low. This is not the same.

We apologize for this oversight. We have updated Figure 1 to read “rhythm check” in place of “pulse check.”

Reviewer #3: The authors address an important problem, ischemia-reperfusion injury, with enhancements to a well-characterized mouse model, cardiac arrest and cardiopulmonary resuscitation (CA/CPR). The difficulty of the established model limits use and replication of results, perhaps impairing the vertical advance of the science. The authors describe several critical enhancements to the model, which it is suggested increase the utility and perhaps generalizability of findings. Improvement in this model, which is critical to investigations in several disciplines, could indeed substantially advance the science of ischemia-reperfusion injury, and consequently have impactful results in the care of patients with critical illness. Therefore this manuscript is appropriate for the journal, and has potential for significant impact. However, some significant concerns should be addressed prior to consideration for publication.

Major concerns:

The major claim of the manuscript is that use of ultrasound to directly deliver potassium to the LV increases access to the model because it reduces the barrier imposed by the difficulty of obtaining IV access in mice. There are two problems with this claim: 1)There is no data to support the claim that ultrasound-guided injection is more easily performed, or more accessible to other labs than IV access followed by IV injection of potassium, and 2)it is at least a little implausible that cardiac ultrasound in the mouse is easier or more accessible to many labs than IV catheter placement given the cost of ultrasound machines and the expertise required to use them in mice. If the authors want to make this claim, they need to provide some data. There are important advantages to this model, and to the technique, but accessibility is unlikely to be the most important one. Therefore, it is suggested that the authors confine the claims made in the conclusion to specific and verifiable claims regarding time, effort, survival, comparability of injury, and time to train, rather than vague claims of accessibility. Other groups have reported the time to prepare a mouse for CA/CPR. This data, if relevant, should be included.

We are grateful to Reviewer #3 for the thoughtful critiques of this manuscript. In response to the reviewer’s comments, we have modified the language in our manuscript to highlight data-driven claims about the advantages of the ultrasound-guided model. First, we have included two references in the Discussion about the length of procedure times. 

The 2nd paragraph of our discussion section has been updated to read: “Our model spares the use of major vessels for drug delivery by using direct LV introduction under ultrasound, resulting in procedure times around 30 minutes from anesthesia induction to extubation, with low surgical mortality (16.7% of SCA mouse surgical mortality, Table 1). While data is limited concerning procedure times in the traditional model, Abella et al. describe 50 minutes of anesthesia induction and up to 40 min of venous instrumentation prior to induction of asystole, followed by 2 hours of invasive monitoring14. More recently, Liu et al. report venous instrumentation followed by 15 min of stabilization, followed by asystole and CPR, and then 30 min of invasive monitoring15. 

We are unable to provide any references concerning preclinical ultrasound adoption or cost. We have included references describing the increased utilization and decreased cost of clinical ultrasound and argue that this adoption will transcend to preclinical work. As suggested by the reviewer, we have removed any claim of accessibility regarding preclinical ultrasound use or claims of barriers to entry from the manuscript.

Discussion, paragraph 3 has been updated to read: “Ultrasound-guided catheter placement has already become a staple of hospital care for many clinicians, allowing physicians and researchers to easily transfer a known skill set into a translational model of SCA16,17. As ultrasound continues to become more affordable and accessible18, we anticipate continued translation of ultrasound reliance into preclinical models. Further, ultrasound utilization allows for continual non-invasive monitoring of cardiac function throughout the arrest, resulting in precise monitoring of cardiac arrest duration. Previously published models that utilize only ECG as an indicator of ROSC may falsely register pulseless electrical activity as a return of circulation, which may erroneously measure ischemic duration. Other models utilize an LV pressure catheter to record the restoration of cardiac flow accurately; however, this requires the placement of an additional invasive cannula. In the current report, the rapid procedure time, high survival, reduced surgical skill required, and venous sparing by this technique are highly advantageous over the traditional model.”

The second claim is that preservation of vessels used for cannulation in other versions of the model reduces procedural times and prevents complications related to these vessels being cannulated or blocked. No evidence is presented that in other models, vessels are blocked, although it’s certainly plausible. Without this evidence, this claim is not verifiable, and like the first claim, is not very important. Similarly, the authors ignore the injury unique to their model, which is 2 30-gauge holes in the left ventricle. Some mention of this, and possible effects, should be made. Were the hearts assessed for injection-related injury at necropsy?

We again thank the reviewer for this comment. We have removed all Discussion concerning complications of venous occlusion from this manuscript. 

We never observed any evidence of coagulated blood in the thorax at the time of necropsy or any trace of pericardial effusion on echocardiography concerning myocardial rupture. We have included language addressing the needle injury in the heart in the discussion section. We’ve amended our Discussion to describe this:

Discussion, paragraph 4: “We did not observe any evidence of pericardial effusion on echocardiography or free blood in the thorax suggestive of myocardial rupture in any of our mice.”

It is not appropriate to report “a trend toward neurologic injury” as a conclusion, as the statistical analysis demonstrates no difference from control. “neurologic injury in some animals” would be an appropriate conclusion. Other authors have reported difficulty obtaining significant neuronal injury at CA times consistent with organism survival, and at least one group has resorted to heating the head while cooling the body (see, eg, Noppens et al, PMID 18957991) to increase neuronal injury while reducing systemic injury. This history would provide context for the finding of minimal neurologic injury in the present model.

We agree with this suggestion and have updated the language in our conclusion to reflect this. The current conclusion paragraph reads:

“This model develops significant cardiac, kidney, and liver injury at one day and neurologic injury in some animals.”

We have modified the paragraph in our discussion section (paragraph 6) concerning neurologic injury to emphasize that while we did some neurologic dysfunction in some mice, we did not see significant changes between sham and SCA groups. We also reference groups that report neurologic changes at prolonged durations of asystole. We suggest that our model could be modified to extend arrest duration, as addressed above in our response to Reviewer #2.

Really, the most important result of this enhancement is the use of intermittent compressions with pulse checks, similar to human CPR. The authors should comment on this.

We thank the reviewer for this comment. We have attempted to emphasize this point in the Discussion when describing the advantages of this model. Discussion, paragraph 3 has been updated to read: “Ultrasound utilization allows for continual non-invasive monitoring of cardiac function throughout the arrest, resulting in the precise assessment of ROSC and duration of asystole by periodic checks, similar to human resuscitation efforts. Previously published models that utilize only ECG as an indicator of ROSC may falsely register pulseless electrical activity as a return of circulation, which may erroneously the time of asystole.”

Minor concerns:

The authors used mice of both sexes, but provide no data on sex differences, which might be important in the overall results. Please provide differential analysis based on sex, or at least denote data points contributed by males vs females in figures.

We agree with this recommendation. We have added supplemental table 1, detailing the surgical characteristics and EF changes between male and female mice in both sham and arrest groups.

Description of temperature dysregulation after CA in mice is not novel, and should be discussed in concert with relevant literature.

In response to comments by Reviewers #1 and #2, we have amended the language in Discussion, paragraph 6, to address the temperature changes observed in our mice and compare this to changes published elsewhere.

How was temperature monitored on an hourly basis in awake mice?

Mice are unconscious or lethargic in the immediate hours following SCA, allowing for monitoring via a rectal probe. We clarified the use of a rectal probe in the methods section.

Were mice reanesthetized for renal ultrasound after ROSC? Were the results from these mice, which may have been affected by additional isoflurane (which has protective effects in ischemia-reperfusion injury) analyzed in the same groups as mice without renal ultrasound?

Anesthesia was discontinued at the time of SCA, which has been clarified in the methods section. 

Following ROSC, mice remain unconscious for approximately one hour, allowing for ultrasound imaging on the kidneys. This point has been amended to the Methods, paragraph 4: “Mice remained unconscious for approximately one hour after SCA, allowing for ultrasound without the need for additional anesthesia.” 

Authors describe this as a model of “sudden cardiac arrest” (SCA), but SCA is really a disease of ventricular tachycardia leading to asystole rather than a disease of asystole due to inability to conduct an impulse (from potassium). Authors should justify the use of the SCA term.

We define sudden cardiac arrest as the sudden cessation of cardiac activity, which our model reflects well. Cardiac arrest has been used by every paper using KCl-induced asystole that we have referenced. We do, however, acknowledge in our limitations section that KCl is an artificial means of inducing SCA that does not recapitulate the arrhythmia seen in human SCA. We have added a limitations section to our paper, noting: “Additionally, KCl is an artificial method for inducing asystole that may not fully recapitulate arrhythmic models of SCA, which may have residual coronary flow and intermittent LV contraction.”

The authors describe anucleated proximal tubular epithelial cells as a sign of nephrotic damage. This is not the correct meaning of the term “nephrotic”, which refers to proteinuria.

We have removed mention of nephrotic disease from the manuscript and instead describe only the histologic and serum changes observed.

The authors report there was no renal blood flow by Doppler until 20 minutes post-ROSC. This is not consistent with the data reported by Hutchens et al (same group as reference 9, PMID 20068453). Reasons for the difference, which would seem important, as the Hutchens paper reports flow measured in the cortex within 5 minutes of ROSC, while the current manuscript reports no flow in the renal artery at that time, would seem important to discuss. It would also be relevant to include discuss relevant or comparable data obtained from humans, if this is available

We apologize for any confusion. We describe 20.55 min of total ischemic time in our model, with the restoration of consistent renal blood flow observed 11.25 minutes after ROSC. The ischemia time is now specified in Discussion, paragraph 7: “By utilizing doppler ultrasonography of the renal artery, we found that the kidney did not receive measurable perfusion until 11.25 minutes after ROSC, for a total mean ischemia duration of 20.55 minutes.”

The Hutchens et al. paper in reference notes restoration of regional renal cortical blood flow measured by local laser doppler at 20% of baseline flow within 5 minutes of ROSC. In their paper, they describe the slow restoration of blood flow over time, terminating at 20 min after ROSC, at which point they report ~50% of the basal flow. We report blood flow through the renal artery but are not able to quantify the percentage of basal perfusion. Our estimates are likely in line with those reported by Hutchens et al. but differ slightly due to differences in the sensitivity of our measurements. We have referenced this work in the discussion section, paragraph 7: “Previous work by Ikeda et al. reports partial restoration of cortical blood flow within minutes of ROSC which gradually improves over the next twenty minutes38, which is in line with our estimates of restoration of renal perfusion by renal artery flow.”

Unfortunately, we were unable to identify any data concerning the duration of renal ischemia in human SCA.

---

## [Decision Letter · Decision Letter 1]

15 Oct 2020

PONE-D-20-22662R1

A novel ultrasound-guided mouse model of sudden cardiac arrest

PLOS ONE

Dear Dr. Kaufman,

Thank you for submitting your manuscript to PLOS ONE. After careful consideration, we feel that it has merit but does not fully meet PLOS ONE’s publication criteria as it currently stands. Therefore, we invite you to submit a revised version of the manuscript that addresses the points raised during the review process.

All the Reviewers and myself believe the previous set of changes greatly improved the manuscript and only Reviewer 3 has two minor comments that should be dealt with.

We look forward to receiving your revised manuscript.

Kind regards,

Daniel M. Johnson, PhD

Academic Editor

PLOS ONE

Reviewers' comments:

Reviewer's Responses to Questions

**Comments to the Author**

1. If the authors have adequately addressed your comments raised in a previous round of review and you feel that this manuscript is now acceptable for publication, you may indicate that here to bypass the “Comments to the Author” section, enter your conflict of interest statement in the “Confidential to Editor” section, and submit your "Accept" recommendation.

Reviewer #1: All comments have been addressed

Reviewer #2: All comments have been addressed

Reviewer #3: (No Response)

2. Is the manuscript technically sound, and do the data support the conclusions?

Reviewer #1: Yes

Reviewer #2: Yes

Reviewer #3: Yes

3. Has the statistical analysis been performed appropriately and rigorously? 

Reviewer #1: Yes

Reviewer #2: Yes

Reviewer #3: Yes

4. Have the authors made all data underlying the findings in their manuscript fully available?

Reviewer #1: Yes

Reviewer #2: Yes

Reviewer #3: No

5. Is the manuscript presented in an intelligible fashion and written in standard English?

Reviewer #1: Yes

Reviewer #2: Yes

Reviewer #3: Yes

6. Review Comments to the Author

Reviewer #1: This revised manuscript is much improved. Methodologies and results are more detailed and robust. I have no further comments.

Reviewer #2: (No Response)

Reviewer #3: This is a very good paper, and the authors have appropriately addressed critique with 2 exceptions:

1. Although some references to "trends toward significance" have been removed, the paragraph including line 314 (in the redline version) continues to use this misleading and erroneous misuse of statistics. Overall, the claim the authors are trying to make here doesn't seem that important (it's a model of global ischemia), so the easiest thing is to just delete this paragraph. But if it's terribly important to make a poorly supported claim about global ischemia, then simply report that some animals had elevated AST/ALT/CPK/LDH, consistent with other CA models and suggesting this is a model of global ischemia, but that larger experiments would be required to fully characterize the model for this role.

2. The third sentence in the conclusions (the one after the edited second sentence) is missing some words and is nonsensical.

7. PLOS authors have the option to publish the peer review history of their article (what does this mean?). If published, this will include your full peer review and any attached files.

Reviewer #1: **Yes: **Andrew Landstrom

Reviewer #2: No

Reviewer #3: No

---

## [Author Response · Author response to Decision Letter 1]

15 Oct 2020

Response to reviewer comments:

We again thank all the reviewers for their thoughtful comments about our work. In the paragraphs below, we have responded to the comments and have indicated the specific changes made to manuscript.

Reviewer #1: This revised manuscript is much improved. Methodologies and results are more detailed and robust. I have no further comments.

We are grateful for this reviewer’s contribution to this manuscript.

Reviewer #2: (No Response)

Reviewer #3: This is a very good paper, and the authors have appropriately addressed critique with 2 exceptions:

1. Although some references to "trends toward significance" have been removed, the paragraph including line 314 (in the redline version) continues to use this misleading and erroneous misuse of statistics. Overall, the claim the authors are trying to make here doesn't seem that important (it's a model of global ischemia), so the easiest thing is to just delete this paragraph. But if it's terribly important to make a poorly supported claim about global ischemia, then simply report that some animals had elevated AST/ALT/CPK/LDH, consistent with other CA models and suggesting this is a model of global ischemia, but that larger experiments would be required to fully characterize the model for this role.

Our apologies for this oversight, we have amended our language as suggested by the author. Paragraph 8 of the Discussion sections has been updated to read “…Some mice had elevated levels of CK, a non-specific marker of muscle degradation, and serum lactate, but these changes were not consistent in the SCA group. While these changes are suggestive of secondary ischemic damage as seen in post-cardiac arrest patients, larger experiments would be necessary to fully characterize these changes in our model.”

2. The third sentence in the conclusions (the one after the edited second sentence) is missing some words and is nonsensical.

Again, we apologize for this error. This sentence has been removed. The updated conclusion paragraph now reads: 

“We demonstrate a novel mouse model of SCA that utilizes direct LV injection of KCl under ultrasound guidance that allows for rapid and reliable arrest with low surgical mortality. This model develops significant cardiac, kidney, and liver injury at one day as well as neurologic injury in some animals. This model lowers the barrier to entry for establishing a mouse model of SCA, which will help researchers investigate the mechanisms underlying SCA mortality.”

---

## [Editor Report · Decision Letter 2]

23 Oct 2020

A novel ultrasound-guided mouse model of sudden cardiac arrest

PONE-D-20-22662R2

Dear Dr. Kaufman,

We’re pleased to inform you that your manuscript has been judged scientifically suitable for publication and will be formally accepted for publication once it meets all outstanding technical requirements.

Kind regards,

Daniel M. Johnson, PhD

Academic Editor

PLOS ONE
---

## [Editor Report · Acceptance letter]

26 Nov 2020

PONE-D-20-22662R2 

A novel ultrasound-guided mouse model of sudden cardiac arrest 

Dear Dr. Kaufman:

I'm pleased to inform you that your manuscript has been deemed suitable for publication in PLOS ONE. Congratulations! Your manuscript is now with our production department. 

Kind regards, 

on behalf of

Dr. Daniel M. Johnson 

Academic Editor

PLOS ONE